# Comprehensive Study of a Diabetes Mellitus Mathematical Model Using Numerical Methods with Stability and Parametric Analysis

**DOI:** 10.3390/ijerph20020939

**Published:** 2023-01-04

**Authors:** Mohammad AlShurbaji, Lamis Abdul Kader, Hadia Hannan, Maruf Mortula, Ghaleb A. Husseini

**Affiliations:** 1Biomedical Engineering Graduate Program, American University of Sharjah, Sharjah P.O. Box 26666, United Arab Emirates; 2Department of Civil Engineering, American University of Sharjah, Sharjah P.O. Box 26666, United Arab Emirates; 3Department of Chemical and Biological Engineering, American University of Sharjah, Sharjah P.O. Box 26666, United Arab Emirates; 4Material Science and Engineering PhD Program, College of Arts and Sciences, American University of Sharjah, Sharjah P.O. Box 26666, United Arab Emirates

**Keywords:** diabetes mellitus, diabetes prevalence, diabetes complications, diabetes control, ODEs, numerical methods, mathematical model, stability analysis

## Abstract

Diabetes is sweeping the world as a silent epidemic, posing a growing threat to public health. Modeling diabetes is an effective method to monitor the increasing prevalence of diabetes and develop cost-effective strategies that control the incidence of diabetes and its complications. This paper focuses on a mathematical model known as the diabetes complication (DC) model. The DC model is analyzed using different numerical methods to monitor the diabetic population over time. This is by analyzing the model using five different numerical methods. Furthermore, the effect of the time step size and the various parameters affecting the diabetic situation is examined. The DC model is dependent on some parameters whose values play a vital role in the convergence of the model. Thus, parametric analysis was implemented and later discussed in this paper. Essentially, the Runge–Kutta (RK) method provides the highest accuracy. Moreover, Adam–Moulton’s method also provides good results. Ultimately, a comprehensive understanding of the development of diabetes complications after diagnosis is provided in this paper. The results can be used to understand how to improve the overall public health of a country, as governments ought to develop effective strategic initiatives for the screening and treatment of diabetes.

## 1. Introduction

Diabetes is sweeping the globe as a silent epidemic, posing a rising threat to public health. On a global scale, there has been a significant increase in the prevalence and incidence of diabetes. According to the most recent 2021 data from the International Diabetes Federation (IDF), 537 million adults, aged 20–79 years, are living with diabetes [1]. This number is estimated to increase to 784 million by 2045. Moreover, diabetes caused 6.7 million deaths in 2021, which is one in every five seconds [1]. Diabetes is the second most prevalent comorbidity in COVID-19, according to epidemiological research [2]. Diabetes patients are more likely to have disease severity, slower recovery, and COVID-19 consequences, including ICU admission, mechanical ventilation, and even death [3]. These issues indicate the dire need for effective intervention techniques and policies critical to the prevention of the rise in the number of individuals with diabetes [1].

Diabetes is a non-communicable, long-term illness characterized by elevated blood sugar levels. Diabetes may affect anyone, irrespective of size, age, or gender. Diabetes can be mainly categorized into two types. Type 1 diabetes is a chronic disease in which the pancreas generates little or no insulin [4]. Type 2 diabetes, the most common type of diabetes, arises when the body grows resistant to insulin or does not produce enough of it [5]. Some of the most common symptoms of diabetes are increased thirst, hunger, fatigue, weight loss, frequent urination, infections, blurry vision, and slow healing of wounds [6]. Several factors might increase the risk of developing the condition, such as physical inactivity, an improper diet, smoking, obesity, stress, increasing age, and family history [5].

If left untreated, diabetes can precede the development of severe complications over time that can harm the heart, blood vessels, eyes, teeth, kidneys, nerves, and ultimately lead to death [7]. Diabetic complications are a leading cause of amputations, which leave the victim permanently disabled. Some of the major complications include cardiovascular disease, which causes mortality among diabetics [1]. Diabetic adults have a two to three times higher risk of heart attacks and strokes [8]. Another complication is diabetic neuropathy, which is defined as nerve damage due to high blood sugar. The most common type is peripheral neuropathy, which mostly affects the feet. When combined with restricted blood flow, it raises the risk of foot ulcers and infections, eventually leading to limb amputations [9]. Furthermore, diabetic retinopathy, or diabetic eye disease, is a common cause of blindness due to long-term damage to the retina’s tiny blood vessels. Diabetes has rendered almost one million individuals blind [10]. Finally, diabetic nephropathy is another major complication. It causes damage to tiny blood arteries in the kidneys and causes kidney disease, which causes the kidneys to become less functional or completely fail. It is one of the most common causes of kidney failure [11].

Diabetes and its complications can be life-threatening and costly. Essentially, effective measures can be taken to help prevent prediabetes and the progression of type 2 diabetes. Maintaining blood glucose, blood pressure levels, and a healthy lifestyle can help delay or prevent diabetes and its complications. The increasing costs of diabetes-related diagnosis, treatment of complications, and management are extremely high. Thus, complications are an essential part of the diabetes epidemiology discussion.

In biomedical sciences, mathematical models and numerical methods have been utilized as theoretical tools for years to study fundamental elements of a wide range of healthcare and biomedical processes, and to develop healthcare strategies [12]. Diabetes modeling is an effective method for countries to monitor the prevalence of diabetes throughout the years and develop cost-effective strategies that control the incidence of diabetes and its complications.

Most of the literature associated with diabetes modeling focuses on glucose and insulin regulatory systems [2,13], diabetes pathways [14], diabetes epidemiology [15,16,17,18,19,20,21], diagnostic test evaluations [18], and the burden of complications [22,23]. The modeling of glucose control in type 2 diabetes, for example, has represented the metabolic problems of this illness using biological mathematical models. In past studies, large mathematical models have been constructed to simulate, examine, and comprehend the dynamics of the diabetes population. For example, both the works of Boutayeb and Chetouani [19] and Derouich et al. [20] provided mathematical models for the dynamics of the diabetic population. Moreover, Nasir and Daud [24] provided a detailed review of different population models of diabetes using ordinary differential equations and their limitations. In addition, Widyaningsih et al. [15] and Kouidere et al. [17,21] created a discrete mathematical model emphasizing the effect of the living environment on diabetes. Likewise, several studies have been conducted on this topic and its related themes. According to the National Diabetes Statistics Report, the incidence of newly diagnosed diabetes among US adults was 5.9 per 1000 people or 1.4 million new cases of diabetes in 2019 [25]. Studies indicate that a significant portion of people with Type 2 diabetes already had diabetic complications in addition to the formation of new diabetes complications at the time of diabetes diagnosis [19,26]. According to Gatwood et al., more than 30% of veterans had chronic kidney disease before being issued a diabetes diagnosis [27].

This paper focuses on a mathematical model simulating the population dynamics of diabetes and its complications. The model was introduced by Boutayeb et al. in 2004, also known as the diabetes complication (DC) model [22]. It is divided into two compartments, diabetics with complications (C) and diabetics without complications (D). In their study, two methods were used to analyze the DC diabetes model: Method I and Euler [5]. Similarly, Akinsola and Oluyo used Euler and Range-Kutta to analyze the DC model [28]. This study used the explicit Euler method, implicit Euler method, Heun’s method, the 4th order Runge–Kutta method, and the Adam-Moulton method to analyze the DC model. The DC model is analyzed using different numerical methods to monitor the size of the diabetic population with and without complications over time. This will provide intriguing opportunities for developing and testing theories, estimating and regulating parameters, comprehending the dynamics of the population, and suggesting practical and effective preventive measures depending on various scenarios [19]. The effect of a growing or declining incidence of diabetes and its complications can be visualized using parameters. The model demonstrates the various strategies that may be developed by varying the parameters that characterize the incidence of diabetes, and diabetes-related comorbidities.

The contributions of this paper are summarized as follows:Investigation of five different numerical methods to analyze the diabetes mathematical model; two have not been used before for this problem. This allows for a comprehensive comparison of the numerical methods and an understanding of the best method to analyze the biological system;Examination of the system behavior for different values of all the parameters, which has not been addressed in the literature. Based on the results, we comment on the best values of the parameters for stability analysis;Investigating the system for different values of diabetes incidence. The incidence of diabetes with complications is significant and of concern, necessitating early and effective strategic initiatives for screening, treatment, and management of diabetes.

## 2. Materials and Methods

Stability, analytical, and numerical methods were used to examine the model. The codes were made using MATLAB software to generate graphs and numerical data. Figure 1 illustrates the DC mathematical model proposed by Akinsola and Temitayo [29]. This is the model that was used in this paper for numerical analysis and to study the control of diabetes epidemiology over the years.

The model examines parameters such as the natural mortality rate (μ), probability of developing a complication (λ), the rate at which complications are controlled (γ), the rate at which patients with complications become severely disabled (v), and mortality rate due to complications (δ). All those parameters affect the model; however, parameters μ, γ, v, and δ are natural rates that describe death or control to patients with diabetes. The only parameter that can be changed is λ, which could be controlled by providing society with motivation to adopt good habits and develop policies such as diet, exercise, regular medical checkups, and more. Therefore, the model examines and analyzes the effect of changing this value that correlated to developing complications, to see its overall effect on the system. The effect of different values of other parameters is also explored. The model is implemented while keeping the incidence of people with diabetes constant. At the time of diagnosis, many patients already have microvascular and macrovascular complications [26], which is reflected in the model. In general, the model’s independent variable is time. In addition, the dependent variables are the number of people who have diabetes without complication *D*(*t*) and the number of people who have diabetes with complication *C*(*t*). *N*(*t*) is the total number of people with diabetes.

### 2.1. ODEs for Modeling Diabetes

The model is explained mainly through two differential equations. One is for diabetics with complications (1) and the other is for non-complication diabetics (2).
(1)$$D^{\prime }\left(t\right)=\frac{dD\left(t\right)}{dt}=I-\left(\lambda +\mu \right)D\left(t\right)+\gamma C\left(t\right)$$
(2)$$C^{\prime }\left(t\right)=\frac{dC\left(t\right)}{dt}=I+\lambda D\left(t\right)-\left(\gamma +\mu +v+\delta \right)C\left(t\right)$$

Formulas (3) and (4) made a first-order system of differential equations where $\theta =\gamma +\mu +v+\delta \;,\;N^{\prime }\left(t\right)=\frac{dN\left(t\right)}{dt}$ and *N*(*t*): is the total number of diabetics (*N* = *C* + *D*). The initial conditions of *C*(*t*) and *N*(*t*) are denoted as *C*_0_ and *N*_0_.
(3)$$C^{\prime }\left(t\right)=I-\left(\lambda +\theta \right)C\left(t\right)+\lambda N\left(t\right)\;,\;\;t>0$$
(4)$$N^{\prime }\left(t\right)=2I-\left(v+\delta \right)C\left(t\right)-\mu N\left(t\right)\;,\;\;t>0$$

Thus, the analytical solutions (5) and (6) were used to find the true value of *C*(*t*) and *N*(*t*). Hence, the true values were used for comparison and computing the percentage error.
(5)$$C\left(t\right)=K_{1}e^{-\eta_{1}t}+K_{2}e^{-\eta_{2}t}+\frac{\alpha }{\beta }I$$
(6)$$N\left(t\right)=K_{1}e^{-\eta_{1}t}+K_{2}e^{-\eta_{2}t}+\frac{\alpha }{\beta }I+\frac{\theta }{\lambda }K_{1}e^{-\eta_{1}t}+\frac{\theta }{\lambda }K_{2}e^{-\eta_{2}t}+\frac{\theta \alpha }{\lambda \beta }I-\frac{I}{\lambda }\;-\frac{1}{\lambda }\left(\eta_{1}K_{1}e^{-\eta_{1}t}\eta_{2}K_{2}e^{-\eta_{2}t}\right)$$
where the parameters of (5) and (6) are defined as follows in (7)–(14):(7)θ=γ+μ+ν+δ
(8)$$\eta_{1}=\frac{1}{2}\left(\sigma -\sqrt[{2}]{\sigma^{2}-4\beta }\right)$$
(9)$$\eta_{2}=\frac{1}{2}\left(\sigma +\sqrt[{2}]{\sigma^{2}-4\beta }\right)$$
(10)σ=ρ+θ+μ
(11)β=ρ(ν+δ)+μ(ρ+θ)
(12)α=2ρ+μ
(13)$$K_{1}=\frac{\beta \left(\rho +\theta -\eta_{2}\right)C_{0}+I\left(\alpha \eta_{2}-\beta \right)-\rho \beta N_{0}}{\beta \left(\eta_{1}-\eta_{2}\right)}$$
(14)$$K_{2}=\frac{-\beta \left(\rho +\theta -\eta_{1}\right)C_{0}+I\left(\beta -\alpha \eta_{1}\right)-\rho \beta N_{0}}{\beta \left(\eta_{1}-\eta_{2}\right)}$$

### 2.2. The Parameters of the ODEs

The parameters of the ODEs play an important role in identifying major changes in diabetic patients over time, with respect to the incidence of diabetes. The critical points of *N*(*t*) and *C*(*t*) as has been found by Akinsola and Temitayo [29] to be as (15) and (16):(15)$$C^{*}\left(t\right)=\frac{\left(2\rho +\mu \right)I}{\rho \left(v+\delta +\mu \right)+\mu \theta }$$
(16)$$N^{*}\left(t\right)=\frac{\left(2\left(\rho +\theta \right)-\left(\mu +\delta \right)\right)I}{\rho \left(v+\delta +\mu \right)+\mu \theta }$$

Then the initial values, as suggested by [22], are (17) and (18):(17)$$C_{0}=C^{*}\pm 500$$
(18)$$N_{0}=N^{*}\pm 500$$

The probabilistic parameters and rates of the ODE model are defined in Table 1. These parameters were used to analyze the numerical solutions of the model. It was observed that most of the authors [19,22,28,30] define γ as the rate at which complications are cured. Similar to Akinsola and Temitayo [29], for ease of comprehension, this paper will refer γ to the rate at which complications are controlled due to the understanding that most of the complications of diabetes are chronic and not curable. The other complications are acute and mostly medical emergency cases [31].

### 2.3. Stability Analysis

If critical points are applied to the system, it reaches a steady state. Through that, we were able to obtain the characteristic equation of the model (21). The quadratic equation is used to find the discriminant (22) and ultimately calculate the eigenvalues (23). Furthermore, the eigenvalues can be analyzed to identify the type of critical point and type of stability of the system. If the determinant is (Δ > 0), then eigenvectors are real and negative. Furthermore, if Δ=0, both eigenvectors equal each other, considered real and negative. Finally, if Δ<0, both eigenvectors are complex conjugates with negative parts. Finally, the type of critical point and stability of the system are shown in Table 2. Using (15)–(18), the eigenvalue is equal to $\chi_{1}$ = −0.156665 and χ_2_ = −0.943351. Furthermore, our Δ = 0.6189. According to Table 2, For the determinant, if (Δ > 0), then the eigenvectors are both real and negative [5]. Furthermore, Table 2 suggests that the model is asymptomatically stable.
(19)$$\;\;\;\;\chi^{2}+\left(\;\lambda +\theta +\mu \right)\chi +\lambda \left(\nu +\delta \right)+\mu \left(\lambda +\theta \right)=0$$
(20)$$\Delta ={\left(\;\lambda +\theta +\mu \right)}^{2}-4\left[\mu \left(\lambda +\theta \right)+\lambda \left(\nu +\delta \right)\right]$$
(21)$$\chi_{1}=\frac{-\left(\lambda +\mu +\delta +\gamma +2v\right)+\Delta^{\frac{1}{2}}}{2}$$
(22)$$\chi_{2}=\frac{-\left(\lambda +\mu +\delta +\gamma +2v\right)-\Delta^{\frac{1}{2}}}{2}$$

### 2.4. Numerical Methods Analysis

Numerical analysis is the process of generating numerical solutions to mathematical expressions using numerical methods [29]. It entails developing, analyzing, and implementing computer systems to solve continuous numerical problems [29]. If the numerical solution approaches the exact solution as the step size approaches zero, the numerical method is convergent [32]. Furthermore, the method must be both convergent and stable because a slight disturbance of the input data does not disrupt the convergence and causes only a minor increase in error [29,33]. The numerical analysis of the model was performed using the MATLAB environment.

The (3) and (4) ODEs were discretized using explicit Euler, implicit Euler, Heun’s, Runge–Kutta (RK) 4th order, and Adam Bashforth–Moulton 4th order methods.

#### 2.4.1. Explicit Euler Method

In the explicit method, the forward finite difference O(h) was used to represent the system of ODEs by obtaining an approximate solution.

Although the explicit method is rather simple, it has the limitation of being conditionally stable. To solve this issue, the amplification factor is suggested to be equal approximately to 1 (when we compare the new value with the one before) to improve the stability of the ODEs in the explicit method and the error as well, since this method has the highest percentage of error out of the other methods.
(23)$$\frac{C_{i}+1-C_{i}}{Dt}=-\alpha C$$
(24)$$\frac{N_{i}+1-N_{i}}{Dt}=-\alpha N$$
where they have a common factor of C_i+1_=−αC_i_Δt + *C*_i_, (similarly applied for *N* as well), thus, the amplification factor (*G*) is equal to:(25)$$G=\frac{C_{i}+1}{C_{i}}=1.007566\approx 1$$

The discretization of Equations (3) and (4) using the Explicit Euler method results in the following numerical equations:(26)$$C_{i+1}=\Delta t\left(I-\left(\lambda +\theta \right)C_{i}+\lambda N_{i}\right)+C_{i}$$
(27)$$N_{i+1}=2\Delta tI-\Delta t\left(v+\delta \right)C_{i}+N_{i}\left(1-\Delta t\mu \right)$$

#### 2.4.2. Implicit Euler Method

The explicit method calculates the future or the new value from the currently known system status. However, the implicit method calculates the future value from the system at present and future times. When the state of *C_i_* and *N_i_* is known, *C_i_*_+1_ and *N_i_*_+1_ can be calculated. Ultimately, the implicit method is unconditionally stable and allows the use of larger step sizes (Δt).

The discretization of Equations (3) and (4) using the implicit Euler method results in the following numerical equations:(28)$$C_{i+1}=\frac{\Delta t\left(I+\lambda N_{i+1}\right)+C_{i}}{1+\Delta t\left(\lambda +\theta \right)}$$
(29)$$N_{i+1}=\frac{\Delta t\left(2I-C_{i+1}\left(\nu +\delta \right)\right)+N_{i}}{1+\Delta t\mu }$$

#### 2.4.3. Heun’s Method

Heun’s method is also another Euler method that is both explicit and implicit. Although it is also based on Euler methods, it provides higher accuracy than both. This method provides an improved accuracy compared to the last two methods due to improving the slope estimation and determination of the two derivatives, one at the beginning of the interval and one at the end. Then, these two derivatives are averaged together to obtain a better slope estimation. Ultimately, Heun’s method is a modified Euler method using the predictor and corrector equations for better results. Heun’s method is a 2^nd^-order error $O\left(h^{2}\right)$ method, which yields better error percentages.

The discretization of Equations (3) and (4) using Heun’s method results in the following numerical equations:(30)$$C_{i+1}^{P}=\;\Delta t\left(I-\left(\lambda +\theta \right)C_{i}+\lambda N_{i}\right)+C_{i}$$
(31)$$N_{i+1}^{P}=\Delta t\left(2I-\left(v+\delta \right)C_{i}-N_{i}\mu \right)+N_{i}$$
(32)$$C_{i+1}^{C}=\frac{\Delta t}{2}\left(\left(I-\left(\lambda +\theta \right)C_{i+1}^{P}+\lambda N_{i+1}^{P}\right)+\left(I-\left(\lambda +\theta \right)C_{i}+\lambda N_{i+1}\right)\right)+C_{i}$$
(33)$$N_{i+1}^{C}=\frac{\Delta t}{2}\left(\left(2I-\left(v+\delta \right)C_{i+1}^{P}-\mu N_{i+1}^{P}\right)+\left(2I-\left(v+\delta \right)C_{i}-\mu N_{i}\right)\right)+N_{i}$$

#### 2.4.4. Runge–Kutta 4th Order Method

The Runge–Kutta (RK) method consists of an ODE that defines $\frac{dC}{dt\;}\;and\;\frac{dN}{dt}$ at initial time *t*(0). The RK method is useful since it is able to find an unknown value of function *t* (time) at any *C* or *N*. The formulas (34)–(37) are used for calculating the next value for *C_i_*_+1_ and N_i+1_ from the previous value *C_i_* and *N_i_*. Δ*C*_1_ represents the slope at the beginning, Δ*C*_2_ represents the slope occurring at the midpoint of the interval between t and Δ*C*_1_, Δ*C*_3_ represents the slope of the midpoint interval between t and Δ*C*_2_, Δ*C*_4_ represents reaching the end of the interval, and similarly for Δ*N*_1,2,3,4_. The RK method is said to be stable if the Eigen values are real without a complex conjugate, less than zero, which is the case in this study.

The discretization of Equations (3) and (4) using the RK 4th method results in the following numerical equations:(34)$$\Delta C_{k}=\Delta t\left(I-\left(\lambda +\theta \right)\left(C_{i}+\frac{\Delta C_{k-1}}{2}\right)+\lambda (N_{i}+\frac{\Delta N_{k-1}}{2})\right)$$
(35)$$\Delta N_{k}=\Delta t\left(2I-\left(v+\delta \right)\left(C_{i}+\frac{\Delta C_{k-1}}{2}\right)+\mu (N_{i}+\frac{\Delta N_{k-1}}{2}\right))$$
where k = 1,2,3,4, and $\Delta C_{0}$ and $\Delta N_{0}$ equal zeros, then:(36)$$C_{i+1}=C_{i}+\frac{1}{6}\left(\Delta C_{1}+2\Delta C_{2}+2\Delta C_{3}+\Delta C_{4}\right)$$
(37)$$N_{i+1}=N_{i}+\frac{1}{6}\left(\Delta N_{1}+2\Delta N_{2}+2\Delta N_{3}+\Delta N_{4}\right)$$

#### 2.4.5. Adam–Moulton Method

The Adam–Moulton method of the 4th order allows us to explicitly compute the approximate solution at an instant time from the solution in previous instants. As the initial value at the initial time step is known, then the RK method can be used to get the midpoints at the intervals (*C*_1_, *C*_2_, *C*_3_, and *C*_4_), and apply them to the predictor and corrector formulas of Adam–Moulton. The predictor formula is used to calculate a rough approximation of *C_i_* and *N_i_* from the RK method, and then approximate the solution by using the corrector formula.

The RK method is used to find *C*_k_ and *N*_k_ when k = 0,1,2,3:(38)$$C_{k}^{\prime }=I-\left(\;\lambda +\theta \right)C_{k}+\lambda N_{k}$$
(39)$$N_{k}^{\prime }=\Delta t\left(2I-\left(v+\delta \right)\;C_{k}-\mu N_{k}\right)$$

Then the predictor and corrector equations for k = 4:(40)$$C_{4}^{P}=C_{3}+\frac{\Delta t}{24}\left(55C_{3}^{\prime }-59C_{2}^{\prime }+37C_{1}^{\prime }-9C_{0}^{\prime }\right)$$
(41)$$N_{4}^{P}=N_{3}+\frac{\Delta t}{24}\left(55N_{3}^{\prime }-59N_{2}^{\prime }+37N_{1}^{\prime }-9N_{0}^{\prime }\right)$$
(42)$$C_{4}^{\prime }=I-\left(\;\lambda +\theta \right)C_{4}^{P}+\lambda N_{4}^{P}$$
(43)$$N_{4}^{\prime }=\Delta t\left(2I-\left(v+\delta \right)C_{4}^{P}-\mu N_{4}^{P}\right)$$
(44)$$C_{4}^{C}=C_{3}+\frac{\Delta t}{24}\left(9C_{4}^{\prime }+19C_{3}^{\prime }-5C_{2}^{\prime }+C_{1}^{\prime }\right)$$
(45)$$N_{4}^{C}=N_{3}+\frac{\Delta t}{24}\left(9N_{4}^{\prime }+19N_{3}^{\prime }-5N_{2}^{\prime }+N_{1}^{\prime }\right)$$

## 3. Results

We implemented the five methods and compared the results with the true values. By calculating the error, the accuracy of the proposed model for the methods used can be evaluated. As real changes in the prevalence will require years to be noted, it is recommended to begin calculating the prevalence after five years of the current time. In addition, the time step (Δt) has been chosen to be one year. Therefore, once the prevalence is noted, the government can plan properly and act quickly on implementing different measures for the sake of the population.

Table 3 shows the results for *C_i_* and *N_i_* using the explicit Euler, implicit Euler, Heun’s, Adam–Moulton, and RK methods from five years and above with Δt=1 year. Noting that the years that we chose to include are only the years 5, 10, 15, and 20, as significant changes can be noticed only every couple of years. The table shows that the RK method is more efficient than the others for this mathematical model.

### 3.1. Stability Analysis

The following sections will provide a comprehensive analysis of the effects of varying the time step size and the parameters of the model on the stability of the system.

#### 3.1.1. The Effect of Varying the Time Step Size (Δt)

The changes in Δt affect the accuracy of the results. Therefore, the lower the value of Δt, the better the results. Table 4 compares the results of the error rate between Δt=1 and Δt=0.5, for all the methods for *C_i_* only, while Table 5 compares the results of *N_i_*. It can be seen that the error rates are much lower for Δt= 0.5, resulting in better accuracy rates of the model.

As the stability is hugely affected by the step size (Δt), this paper aims to study the stability of all the methods used while changing the Δt. The values of Δt used in this paper range from 0. 1 to 1.5. Figure 2 shows the results for both *C_i_* and *N_i_* for (a) explicit Euler, (b) implicit Euler, (c) Heun’s, (d) Adam–Moulton, and (e) RK methods. It can be seen that some methods are diverging, beginning from Δt=1.5 and above. In addition, the results with smaller Δt are much more accurate and stable.

#### 3.1.2. The Effect of Varying the Parameters Values

The parameters of the system defined in Table 1 are specific values that ensure the stability of the system. In addition, the stability of the model using the RK method was evaluated while altering the parameters mentioned in Table 1, which are γ,λ,δ,μ, and v.

##### The Rate at Which Complications Are Controlled (γ)

The investigated γ values in this paper are 0.04, 0.08, 0.5, and 1.0. These values were obtained from [22]. Figure 3 shows the results of *C_i_* and *N_i_* for the RK method in terms of γ and Δt, where (a) γ=0.04, (b) γ=0.08, (c) γ=0.5, and (d) γ=1.0. It can be seen that as larger the γ value, the larger the error rates. After γ=1.0, the results started to diverge.

##### Probability of Developing a Complication (λ)

The investigated λ values in this paper are 0.04, 0.66, 0.85, and 1.2. Figure 4 shows the results of *C_i_* and *N_i_* for the RK method in terms of λ and Δt, where (a) λ=0.04, (b) λ=0.66, and (c) λ=0.85.

##### The Mortality Rate Due to Complications (δ)

The investigated δ values in this paper are 0.02, 0.05, 0.1, and 0.35. Figure 5 shows the results of C_i_ and N_i_ for the RK method in terms of δ and Δt, where (a) δ=0.02, (b) δ=0.05, (c) δ=0.1, and (d) δ=0.35.

##### Natural Mortality Rate (μ)

The investigated μ values in this paper were 0.005, 0.02, and 0.45. Figure 6 shows the results of *C_i_* and *N_i_* for the RK method in terms of μ and Δt, where (a) μ=0.005, (b) μ=0.02, (c) and μ=0.45.

##### The Rate at Which Patients with Complications Become Severely Disabled (v)

The investigated v values in this paper are 0.03, 0.05, 0.1, and 0.35. Figure 7 shows the results of *C_i_* and *N_i_* for the RK method in terms of v and Δt, where (a) v=0.03, (b) v=0.05, (c) v=0.1, and (d) v=0.35.

## 4. Discussion

The mathematical modeling of diabetes aids in the analysis and interpretation of population changes. In previous studies, some papers have solved the diabetic model using parameters related to the specific country under study. This paper adds to the literature by performing a comprehensive evaluation of the effect of the time step and all the possible values of relevant parameters using multiple methods that can be applied to any country.

In this study, two ODEs were to analyze the development of diabetic patient numbers with complications. The ODEs were solved using five numerical methods, namely, explicit Euler, implicit Euler, Heun’s, RK (4th order), and Adam–Moulton (4th order). The implicit, explicit, and RK methods were used in previous studies, and in this paper, Heun’s, and Adam–Moulton’s methods are presented for the first time to study and analyze the prevalence of diabetes. After discretization and implementation of all methods, the RK method resulted in the best accuracy rates, with a low $1\times 10^{-6}\;$% error rate. RK method is compared to other implemented methods in this paper due to its superior performance. particularly, the performances of the RK and the Adam–Moulton methods are always compared for first-order ODEs. According to Gofe and Gebregiorgis [30], the Adam–Moulton method is considered the most efficient due to the predictor and modifier equations in the model. However, in terms of accuracy, there is no generalization on the best method, and it depends solely on the ODEs used. As a result, it is recommended to observe the performance of each method based on the performance of ODE after doing the stability analysis. Moreover, in the literature, the implicit Euler method showed a smaller percentage error than the explicit Euler. This paper highlights similar results [23]. Also, the Implicit method proved more accurate than Heun’s method for the problem at hand.

It can be noticed that the model is stable using implicit Euler, Heun’s, and RK’s methods, as they perform and calculate implicitly. The explicit Euler and Adam–Moulton’s methods are stable for Δt less than 1.0 and begin to become unstable for Δt values larger than one year, as shown in Figure 2; this is because both work explicitly. When Δt = 1.5, the model shows a damping error as shown in Figure 2a for the explicit method. Although Adam–Moulton’s method is less accurate than the RK method, it still shows good stability analysis, as indicated in Figure 2e for values of Δt ranging from 0–0.5. Thus, the Adam method is suitable for those who require frequent examination of the prevalence of diabetes to monitor minor changes. Note that the nature of Adam–Moulton’s method depends on four previous values, as shown in Equations (40)–(45), to calculate the current value. Therefore, the values of the stability analysis of Adam–Moulton’s method start after four steps.

Furthermore, the stability analysis of λ, γ, and more parameters in the diabetes model for the RK method was investigated. Figure 3 shows the analysis of the parameter γ (the rate of control of complications) which ranged between 0.04 and 1. it can be visualized that as the rate of complications increases, the number of diabetic patients with complications decreases. Hence, when gamma equals zero, meaning the complications are uncontrollable, death of diabetic people will occur due to the rise of diabetic patients with complications. However, if the rate of controlling the complications is slightly controlled, at a rate of 0.5, a substantial effect will be seen on the value of *C*. Moreover, for γ = 0.04, the value of *C* is more evened out. Ultimately, complications are always on the rise for diabetic patients since there is no permanent cure.

Consequently, Figure 4 shows the analysis for λ. The higher the probability of developing complications, the higher the number of diabetics with complications. It can be concluded from Figure 4, that the lower the probability of developing complications, diabetic patients without complications will increase. In developed countries, diabetes is an epidemic with most cases without complications. This is due to the low rate of developing complications and a high rate of controlling the complications through laws implemented by public health authorities [30]. Ultimately, our model shows consistent findings of the parametric analysis with Akinsola et al., who conducted a similar investigation [29].

In addition, the mortality of diabetic patients arises from the complications associated with the disease. In the DC model of diabetes, there are two parameters that are concerned with the mortality of diabetic patients with complications: the natural mortality rate (μ) and the mortality rate due to complications (δ). Figure 5 shows that the mortality rate due to complications increases, which corresponds to statistically more death cases of diabetic patients. Thus, the overall prevalence of diabetes in society decreases. Similarly, as the natural mortality rate increases in Figure 6, the value of C decreases.

Another parameter (ν) was analyzed using the RK method, as shown in Figure 7. This parameter is the rate at which patients with complications became severely disabled. People with diabetes-related disabilities [34] are at a higher risk of mortality as they are unable to control and manage their condition effectively. Therefore, this will increase the mortality rate due to diabetic complications, and eventually decrease the value of C. Care must be taken to reduce this rate by providing management programs to assist and support people in this category.

In Figure 8, the true value of *C_i_* is plotted in blue. However, it is not shown because it is very close to the values of the RK method. Thus, the graph distinctly illustrates that RK has the highest accuracy, followed by Adam–Moulton’s method.

Ultimately, for the RK method, the *C* values do not vary for different values of gamma, and this is due to the high accuracy of this method for the suggested ODE system. On the other hand, the Adam–Moulton method shows *C* values that are oscillating at γ equal to 0.5. This value was recommended by Boutayeb’s paper [14]. However, it does not fit the model since it causes unstable results for the Adam–Moulton method. Hence, this paper proposes using γ to be equal to 0.08, which provides good stability for the Adam–Moulton method and the rest of the methods.

For further analysis, the incidence of diabetes was considered in this paper. Three levels of incidence of diabetes are considered: low, medium, and high levels of incidence. Additionally, three different values of the parameters λ are studied for the various levels of incidence. By varying the parameter values, nine different events may be taken into consideration as can be seen in Table 6. For example, the scenario “high–high” denotes both a high rate of complications and a high incidence of diabetes.

From the above table, it can be noticed that there is an increase in both the longitudinal and latitudinal directions of the table. This indicates that the number of diabetics with complications and the total number of people with diabetes increase as the incidence of diabetes increases and as the rate of developing complications increases. This suggests that without effective health strategies and policies in place, the diabetes population may surge to a level beyond control, posing a threat to the resources of the country. In the situation of uncontrolled diabetes, the health authorities may not be able to serve or provide care for all the affected people, causing a significant threat to the socioeconomic welfare of a country.

The burden of diabetes can be significantly reduced if proper measures and schemes are employed at different levels to check the increase in diabetes incidence. First, by lowering the number of individuals who acquire prediabetes and individuals who get diabetes without complications, be it through diabetes education and health policies to promote good health and raise awareness, especially among the youth. There is a strong probability of reversing the effects of diabetes in its early stages. Secondly, by lowering the number of individuals with diabetes who have complications. This action has two main implications. Early detection of diabetes requires preventative measures, and after someone has been diagnosed with diabetes, care and precaution should be taken to prevent or, at the very least, delay the development of complications with the available resources [29].

Therefore, if the incidence is reduced, the number of pre-diabetics, diabetics, and diabetics with complications will also decrease. This goal can be realized through strategies that focus on raising awareness about a healthy diet, encouraging physical exercise, managing obesity, addressing blood pressure, reducing stress levels, quitting smoking and bad drinking habits, and regular screening.

We noted the ambiguity and dearth of data on the incidence of diabetes with complications and the worldwide status of diabetes complications rates. This gap in data on trends in complications has hindered the effective evaluation of the health situation of a country [35]. Future research on global trends in diabetes complications is necessary for better resource management and the identification of underlying factors and risks faced by the population.

The mathematical model illustration supports the diagnoses and recommendations of diabetes specialists and healthcare managers in general. Moreover, the results of our model are concurrent with the results from literation on similar models [19,22,29]. Thus, it provides health decision-makers with a framework for comparing the socioeconomic consequences of uncontrolled diabetes, and an effective strategy is a better investment in healthcare to reduce costs in the long run [19].

## 5. Conclusions

Diabetes and its complications pose an increasing threat to healthcare systems across the world. In this paper, a mathematical model of diabetes and the possible complications in a population was investigated and numerically analyzed. This paper added more numerical methods than were previously used. New methods such as the Heun’s and the Adam–Moulton fourth-order methods were added. In addition, the step size (Δt) and its effect on the system were evaluated by computing the percentage error at every time-step size. The stability, numerical analysis, and simulations were determined for various stated values of the parameters for the analysis of the model. According to the results provided in the paper, the RK fourth-order method had the highest accuracy when compared with the other methods. Furthermore, by analyzing the parameters based on different variations, different situations can be generated, and subsequent strategies can be used in accordance with the available resource options. The optimal parameters’ values were recommended to get the best results. From the graphs, if the incidence of diabetes is controlled, then the number of diabetics with or without complications also reduces.

In conclusion, controlling the rate of developing diabetes and monitoring complications in real life can be achieved through health education, healthy diet awareness, regular exercise, quitting smoking, and reducing other metabolic risks like obesity and hypertension. These actions will help in implementing optimal strategies to reduce the incidence of diabetes and the total number of diabetic patients. The numerical analysis of the model affirms the facts of rising diabetes incidence and prevalence around the world, according to IDF statistics. This emphasizes the significance of early diagnosis, monitoring, and treatment of diabetes mellitus to minimize complications, as well as provide diabetes patients with sufficient medical care. Special consideration should be given to policies and strategies that promote awareness of good health and the benefits of the prevention of diabetes rather than focusing primarily on curing the sick.

In the future work of this project, the partial differential equations can be investigated for more specific modeling of diabetes. Moreover, there should be more research with a special emphasis on finding the incidence of diabetes with and without complications. Research to find the incidence of diabetes showed that most international organizations do not report on the incidence of diabetes with complications. It is important to report this number for a better assessment of the health status of a country. To conclude, this paper can be improved by adding more mathematical methods to compare and make an affirmative conclusion in which method is highly recommended. Moreover, real-life diabetic data acquired from the public to be utilized for further testing of the model and provide attestation to our results. Lastly, if habits, health education and awareness, and diet are changed over the years, the results obtained from the model might differ significantly. the aforementioned points are considered as limitations in this paper.

## Figures and Tables

**Figure 1 ijerph-20-00939-f001:**
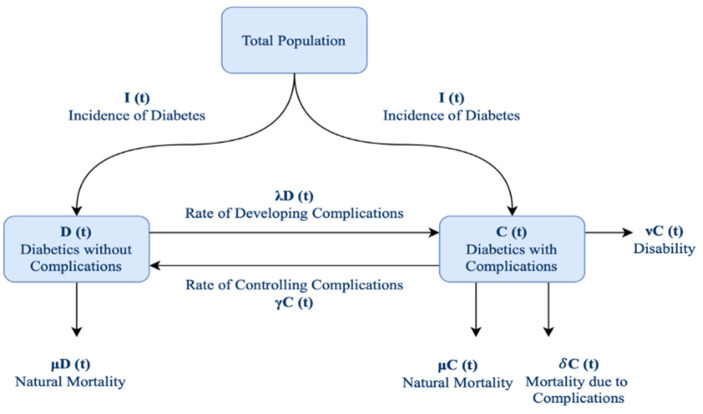
The DC model.

**Figure 2 ijerph-20-00939-f002:**
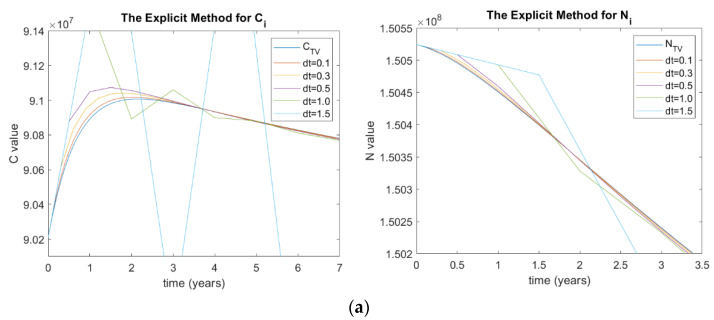
(**a**) Stability analysis of *C_i_* and *N_i_* in terms of ∆t for implicit Method; (**b**) for explicit method; (**c**) Heun’s method; (**d**) RK method; (**e**) Adam’s method, where C is the number of diabetic patients with complications and $C_{TV}$ is its true value, N  is the total number of people with diabetes and $N_{TV}$ is its true value, and dt is the time step size.

**Figure 3 ijerph-20-00939-f003:**
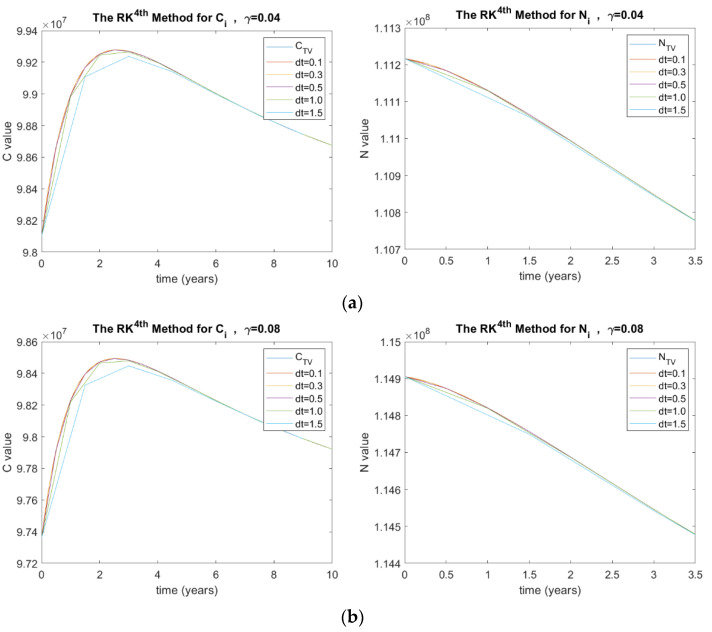
(**a**) Stability analysis for RK method when γ=0.04, (**b**) γ=0.08, (**c**) γ=0.5, and (**d**) γ=1.0, where C  is the number of diabetic patients with complications and $C_{TV}$ is its true value, N  is the total number of people with diabetes and $N_{TV}$ is its true value, and dt is the time step size.

**Figure 4 ijerph-20-00939-f004:**
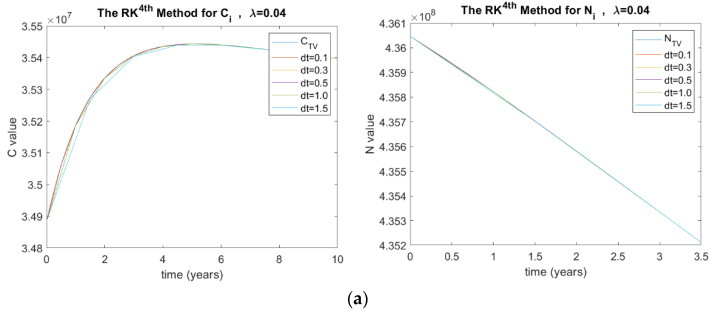
(**a**) Stability analysis for RK method when λ=0.04, (**b**) λ=0.66, and (**c**) λ=0.85. Where C  is the number of diabetic patients with complications and $C_{TV}$ is its true value, N  is the total number of people with diabetes and $N_{TV}$ is its true value, and dt is the time step size.

**Figure 5 ijerph-20-00939-f005:**
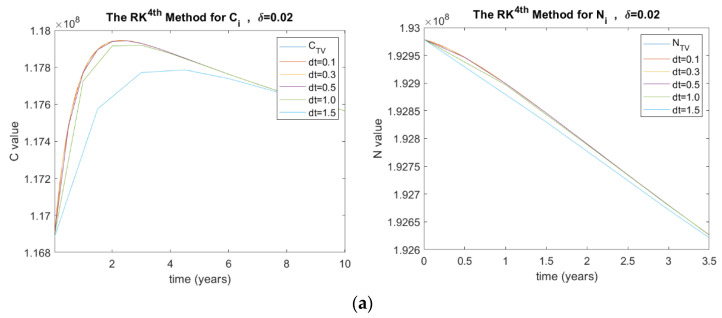
(**a**) Stability analysis for RK method when δ=0.02, (**b**) δ=0.05, (**c**) δ=0.1, and (**d**) δ=0.35, where C  is the number of diabetic patients with complications and $C_{TV}$ is its true value, N  is the total number of people with diabetes and $N_{TV}$ is its true value, and dt is the time step size.

**Figure 6 ijerph-20-00939-f006:**
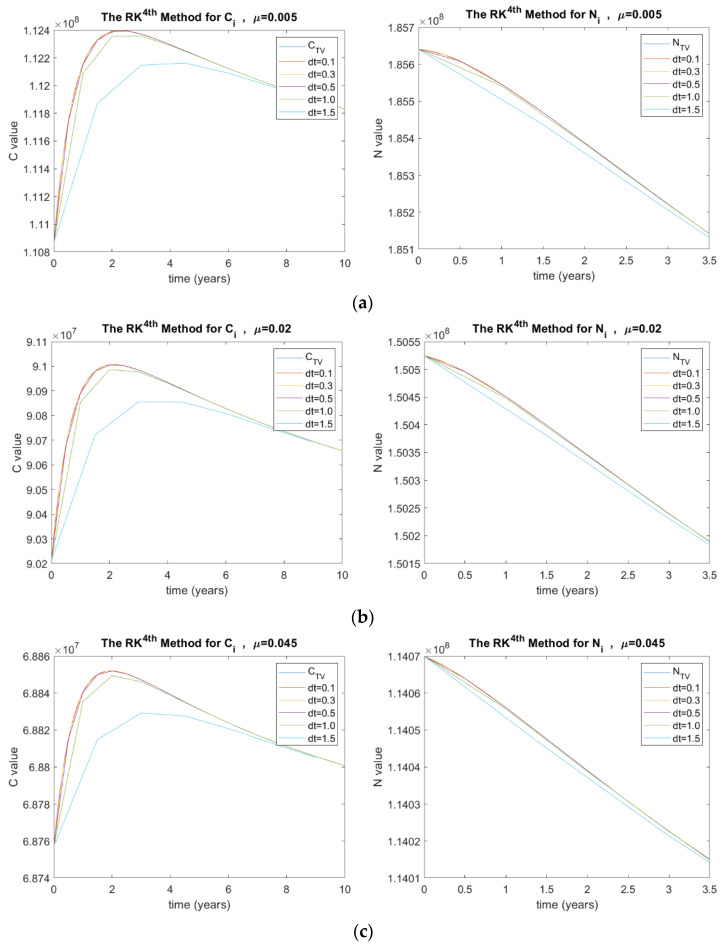
(**a**) Stability analysis for the RK method when μ=0.005, (**b**) μ=0.02, (**c**) and μ=0.45, where C  is the number of diabetic patients with complications and $C_{TV}$ is its true value, N  is the total number of people with diabetes and $N_{TV}$ is its true value, and dt is the time step size.

**Figure 7 ijerph-20-00939-f007:**
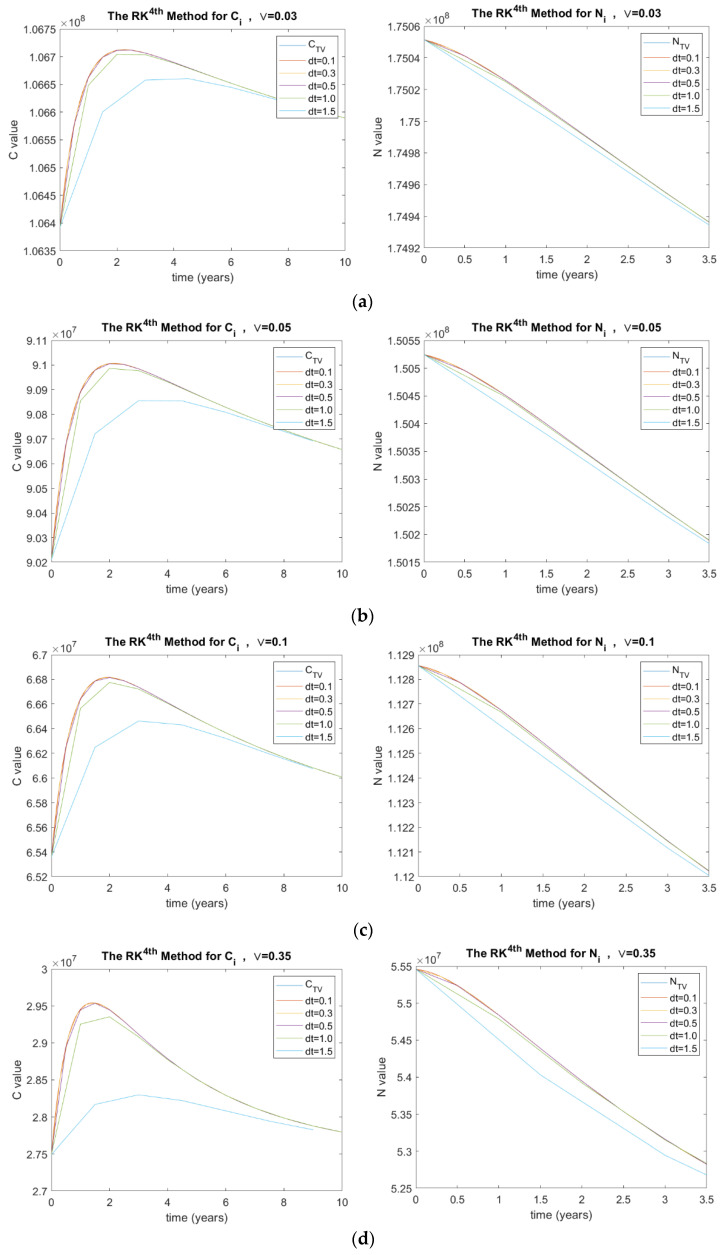
(**a**) Stability analysis for RK method when v=0.03, (**b**) v=0.05, (**c**) v=0.1, and (**d**) v=0.35, where C  is the number of diabetic patients with complications and $C_{TV}$ is its true value, N  is the total number of people with diabetes and $N_{TV}$ is its true value, and dt is the time step size.

**Figure 8 ijerph-20-00939-f008:**
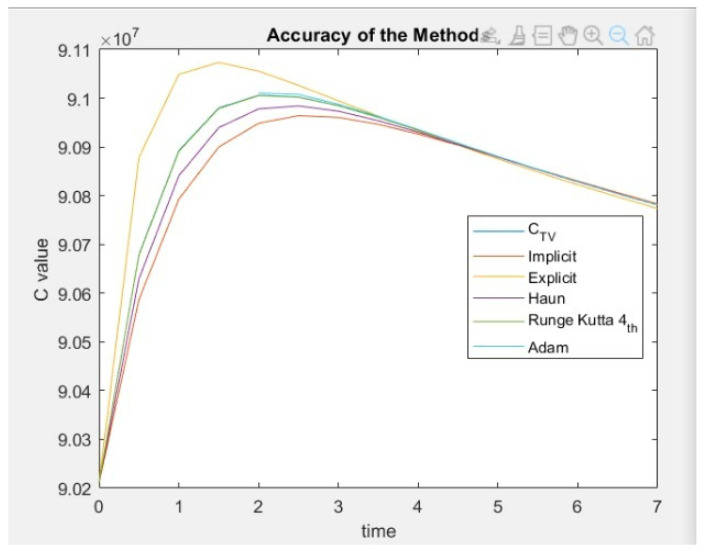
Comparison between the results of all the methods, where C  is the number of diabetic patients with complications and $C_{TV}$ is its true value.

**Table 1 ijerph-20-00939-t001:** Parameter description and values used in discretization of ODE.

Parameter	Definition	Value
*C*(*t*)	Number of diabetic patients with complications	Determined by the method used after discretization of ODE
*D*(*t*)	Number of diabetic patients without complications	Determined by the method used after discretization of ODE
*N*(*t*)	Total number of people with diabetes	Determined by the method used after discretization of ODE
λ	Probability of developing a complication	0.85
	The mortality rate due to complications	0.05
μ	Natural mortality rate	0.02
γ	The rate at which complications are controlled	0.5
v	The rate at which patients with complications become severely disabled	0.05
I	Incidence of diabetes	6 × 10^6^

**Table 2 ijerph-20-00939-t002:** Type of critical points and stability based on the eigenvalues and determinants [5].

Eigenvalues	Type of Critical Point	Stability
$\chi_{1}$ > $\chi_{2}$ > 0	Node	Unstable
$\chi_{1}$ < $\chi_{2}$ < 0	Node	Asymptomatically stable
$\chi_{2}$ < 0 < $\chi_{1}$	Saddle point	Unstable
$\chi_{1}$ = $\chi_{2}$ > 0	Proper or improper node	Unstable
$\chi_{1}$ = $\chi_{2}$ < 0	Proper or improper node	Asymptomatically stable
$\chi_{1}$ = $\chi_{2}$ = λ±iμ	Spital point	Unstable
$\chi_{1}$ = iμ $\chi_{2}$ = −iμ	Center	Asymptomatically stable

$\chi_{1}\;$and $\chi_{2}$ are the eigenvalues, λ is the probability of developing a complication, and μ is the natural mortality rate.

**Table 3 ijerph-20-00939-t003:** The results using the five methods.

*t* (Year)	*C_i_*	*C_i_* (True Value)	Error (*C_i_*)	*N_i_*	*N_i_* (True Value)	Error (*N_i_*)
**Explicit Euler**						
5	90,880,914.61	90,880,136.51	0.0009%	150,029,621.02	150047685.76	0.0120%
10	90,641,567.00	90,657,024.59	0.0171%	149,656,724.42	149681776.36	0.0167%
15	90,492,596.39	90,507,781.43	0.0168%	149,413,117.27	149437930.11	0.0166%
20	90,394,964.97	90,408,340.72	0.0148%	149,253,601.86	149275456.17	0.0146%
Avg.			0.0124%			0. 0150%
**Implicit Euler**						
5	90,879,176.47	90,880,136.51	0.0011%	150,064,146.12	150,047,685.76	0.0110%
10	90,671,086.35	90,657,024.59	0.0155%	149,704,989.60	149,681,776.36	0.0155%
15	90,522,098.11	90,507,781.43	0.0158%	149,461,324.77	149,437,930.11	0.0157%
20	90,421,162.05	90,408,340.72	0.0142%	149,296,404.69	149,275,456.17	0.0140%
Avg.			0.0117%			0.0140%
**Heun’s**						
5	90,813,160.25	90,880,136.51	0.0737%	150,043,360.49	150,047,685.76	0.0029%
10	90,652,836.25	90,657,024.59	0.0046%	149,682,137.68	149,681,776.36	0.0002%
15	90,507,893.51	90,507,781.43	0.0001%	149,438,600.88	149,437,930.11	0.0004%
20	90,408,696.63	90,408,340.72	0.0004%	149,276,070.68	149,275,456.17	0.0004%
Avg.			0. 0197%			0.00098%
**Adam-Moulton (4th order)**						
5	90,983,291.97	90,880,136.51	0.1135%	150,055,113.24	150,047,685.76	0.0049%
10	90,659,713.29	90,657,024.59	0.0030%	149,681,969.31	149,681,776.36	0.0001%
15	90,363,636.94	90,507,781.43	0.1593%	149,427,550.24	149,437,930.11	0.0069%
20	90,259,489.65	90,408,340.72	0.1646%	149,264,737.44	149,275,456.17	0.0072%
Avg.			0.1101%			0.0048%
**RK (4th order)**						
5	90,879,229.41	90,880,136.51	0.00099813%	150,047,620.61	150,047,685.76	0.00004342%
10	90,657,022.33	90,657,024.59	0.00000249%	149,681,776.42	149,681,776.36	0.00000004%
15	90,507,781.57	90,507,781.43	0.00000015%	149,437,930.34	149,437,930.11	0.00000015%
20	90,408,340.85	90,408,340.72	0.00000014%	149,275,456.38	149,275,456.17	0.00000014%
Avg.			0.00025023%			0.00001094%

$C_{i}$ is the number of diabetic patients with complications, $N_{i}\;$ is the total number of people with diabetes, *t* is the time in years, and Error() is the difference between $C_{i}$ or $N_{i}$ and their true values.

**Table 4 ijerph-20-00939-t004:** The effect of changing Δt on the results of *C_i_*.

***t* (Year)**	**Explicit**		Implicit		Heun’s		Adam-Moulton		RK	
	E(*C_i_*) at Δt = 1	E(*C_i_*) at Δt = 0.5	E(*C_i_*) at Δt = 1	E(*C_i_*) at Δt = 0.5	E(*C_i_*) at Δt = 1	E(*C_i_*) at Δt = 0.5	E(*C_i_*) at Δt = 1	E(*C_i_*) at Δt = 0.5	E(*C_i_*) at Δt = 1	E(*C_i_*) at Δt = 0.5
5	0.0009%	0.0053%	0.0011%	0.0016%	0.0737%	0.001449%	0.1135%	0.00045917%	0.00099813%	0.00002555%
10	0.0171%	0.0083%	0.0155%	0.0080%	0.0046%	0.000109%	0.0030%	0.00000053%	0.00000249%	0.00000004%
15	0.0168%	0.0083%	0.0158%	0.0080%	0.0001%	0.000114%	0.1593%	0.00000003%	0.00000015%	0.00000001%
20	0.0148%	0.0073%	0.0142%	0.0072%	0.0004%	0.000101%	0.1646%	0.00000003%	0.00000014%	0.00000001%
Avg.	0.0124%	0.0073%	0.0117%	0.0062%	0. 0197%	0.000430%	0.1101%	0.00014940%	0.00025023%	0.00000640%

$C_{i}$ is the number of diabetic patients with complications, E($C_{i}$) is the difference between $C_{i}$ and its true value, *t* is the time in years, and Δt is the time step size.

**Table 5 ijerph-20-00939-t005:** The effect of changing Δt on the results of *N_i_*.

***t* (Year)**	**Explicit**		**Implicit**		**Heun’s**		**Adam-Moulton**		**RK**	
	E(*N_i_*) at Δt = 1	E(*N_i_*) at Δt = 0.5	E(*N_i_*) at Δt = 1	E(*N_i_*) at Δt = 0.5	E(*N_i_*) at Δt = 1	E(*N_i_*) at Δt = 0.5	E(*N_i_*) at Δt = 1	E(*N_i_*) at Δt = 0.5	E(*N_i_*) at Δt = 1	E(*N_i_*) at Δt = 0.5
5	0.0120%	0.0061%	0.0110%	0.0057%	0.0029%	0.000017%	0.0049%	0.00002001%	0.00004342%	0.00000111%
10	0.0167%	0.0082%	0.0155%	0.0079%	0.0002%	0.000112%	0.0001%	0.00000001%	0.00000004%	0.00000001%
15	0.0166%	0.0082%	0.0157%	0.0079%	0.0004%	0.000001%	0.0069%	0.00000003%	0.00000015%	0.00000001%
20	0.0146%	0.0072%	0.0140%	0.0071%	0.0004%	0.000101%	0.0072%	0.00000003%	0.00000014%	0.00000001%
Avg.	0. 0150%	0.00743%	0.0140%	0.0072%	0.0010%	0.000058%	0.0048%	0.00000502%	0.00001094%	0.00000029%

$C_{i}\;$is the number of diabetic patients with complications, E($C_{i}$) is the difference between $C_{i}$ and its true value, *t* is the time in years, and Δt is the time step size.

**Table 6 ijerph-20-00939-t006:** The effect of changing incidence of diabetes and parameter λ on the results of C and N.

	*I*		Low (2 × 10^6^)	Medium (3 × 10^6^)	High (6 × 10^6^)
*λ*					
**Low (0.3)**		C	25,856,897.3	38,785,345.96	77,570,691.9
		N	72,685,790.24	109,028,685.4	21,805,7371.0
**Medium (0.4)**		C	27,369,403.59	41,054,105.38	82,108,210.8
		N	64,835,059.16	97,252,588.74	194,505,177.0
**High (0.5)**		C	28,374,387.21	42,561,580.82	85,123,161.6
		N	59,592,864.31	89,389,296.47	178,778,593.0

C is the number of diabetic patients with complications, N is the total number of people with diabetes, λ is the probability of developing a complication, and I is the incidence of diabetes.

## Data Availability

Not applicable.

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
