# Peer review of "Comprehensive Study of a Diabetes Mellitus Mathematical Model Using Numerical Methods with Stability and Parametric Analysis"

_ijerph, 2023, doi:10.3390/ijerph20020939_

Round 1

Reviewer 1 Report

The study by Alshrbaji et al. is well-designed and interesting. The results are clearly presented and methods are well described. However, the introduction and discussion parts are relatively long and lack focus, they need to be condensed. The conclusion is well supported by he presented data, however, is stretched and still needs more focus. 

Author Response

Please find our attached response

Reviewer 2 Report

This article addresses an imperative topic, especially given the increasing incidence of diabetes mellitus and its chronic complications. Moreover, these numbers represent a considerable burden for public health, being a real challenge for most of the health care systems globally speaking.

The title describes the study in an intelligible manner, while the concise abstract provides suitable data on the idea of the study.

The introduction presents in a thorough manner the scientific background, with adequate information on diabetes epidemiology, and gives us the specific objectives of the study.

Speaking of Materials and Methods, I would recommend you mention the name of the author of the DC mathematical model, not just cite the study (page 4, line 151).

I am interested in knowing your opinion about your study limitations, besides the paucity of data you mentioned.

I would advise you to revise reference 13 (accurately write the DOI number).

To sum up, this article would make an important contribution to scientific literature, proposing a mathematical method to assess the incidence of diabetes, as well its complications and the impact of these on the health care system. Additionally, your article brings to the table comparable methods, explaining the applicability of each one, for better practice of them in government policies and public health management. I look forward to witnessing these methods' impact on public health protocols and governmental investment decisions.

Author Response

Please find our attached response

Reviewer 3 Report

This article contains updated topics in the etiology of diabetic with special reference to the implication of mathematical model. This is an interesting study suggesting the importance of mathematical models in the quick diagnosis of diabetes. Considering the prevalence of diabetes, this manuscript will attract broad range of readers. I do not have any critical comments.

Minor issues to strengthen this manuscript are raised as follows:

1. Abbreviations used in tables should be explained in the footnote of these tables.

2. In the table titles, it suggests removing the dots.

3. Abbreviations used in figures should be explained in the description of the figures.

4. line 77. I suggest clarifying the definition of diabetic neuropathy. Do  not simply state that the nerve damge in the feet may indicate the pathogenesis of diabetic neuropathy. Which tissues? What kind of changes related? What kind of pathology related? Tt should be noted that although all types of neuropathy can be seen in diabetes, the most common is symmetrical axonal (length-dependent) neuropath. 

Author Response

Please find our attached response.
